# Coordination Behavior of 1,4-Disubstituted Cyclen Endowed with Phosphonate, Phosphonate Monoethylester, and H-Phosphinate Pendant Arms

**DOI:** 10.3390/molecules24183324

**Published:** 2019-09-12

**Authors:** Jiří Bárta, Petr Hermann, Jan Kotek

**Affiliations:** Department of Inorganic Chemistry, Faculty of Science, Charles University, Hlavova 8, 128 43 Prague 2, Czech Republic; BJRS@seznam.cz (J.B.); petrh@natur.cuni.cz (P.H.)

**Keywords:** macrocyclic ligands, metal complexes, manganese, gadolinium, copper, MRI contrast agents, phosphonate ligands, phosphinate ligands, cyclen derivatives, protonation constants, stability constants

## Abstract

Three 1,4,7,10-tetraazacyclododecane-based ligands disubstituted in 1,4-positions with phosphonic acid, phosphonate monoethyl-ester, and H-phosphinic acid pendant arms, 1,4-H_4_do2p, 1,4-H_2_do2p^OEt^, and 1,4-H_2_Bn_2_do2p^H^, were synthesized and their coordination to selected metal ions, Mg(II), Ca(II), Mn(II), Zn(II), Cu(II), Eu(III), Gd(III), and Tb(III), was investigated. The solid-state structure of the phosphonate ligand, 1,4-H_4_do2p, was determined by single-crystal X-ray diffraction. Protonation constants of the ligands and stability constants of their complexes were obtained by potentiometry, and their values are comparable to those of previously studied analogous 1,7-disubstitued cyclen derivatives. The Gd(III) complex of 1,4-H_4_do2p is ~1 order of magnitude more stable than the Gd(III) complex of the 1,7-analogue, probably due to the disubstituted ethylenediamine-like structural motif in 1,4-H_4_do2p enabling more efficient wrapping of the metal ion. Stability of Gd(III)–1,4-H_2_do2p^OEt^ and Gd(III)–H_2_Bn_2_do2p^H^ complexes is low and the constants cannot be determined due to precipitation of the metal hydroxide. Protonations of the Cu(II), Zn(II), and Gd(III) complexes probably takes place on the coordinated phosphonate groups. Complexes of Mn(II) and alkali-earth metal ions are significantly less stable and are not formed in acidic solutions. Potential presence of water molecule(s) in the coordination spheres of the Mn(II) and Ln(III) complexes was studied by variable-temperature NMR experiments. The Mn(II) complexes of the ligands are not hydrated. The Gd(III)–1,4-H_4_do2p complex undergoes hydration equilibrium between mono- and bis-hydrated species. Presence of two-species equilibrium was confirmed by UV-Vis spectroscopy of the Eu(III)–1,4-H_4_do2p complex and hydration states were also determined by luminescence measurements of the Eu(III)/Tb(III)–1,4-H_4_do2p complexes.

## 1. Introduction

Stable metal chelates belong to a group of intensively studied compounds due to their various applications. Importance of the research has been proven, e.g., by nearly four decades of development of magnetic resonance imaging (MRI) where such paramagnetic complexes are used as contrast agents (CAs). A wide application scope of MRI and its great impact on healthcare have been a driving force behind the MRI CAs development focusing on complexes of Gd(III) and Mn(II) ions [1,2,3,4,5,6,7,8,9,10]. Due to a toxicity of the metal ions (doses of the CAs used for an efficient contrast enhancement are relatively high and exceed a lethal dose of the “free” aqua metal ions), the ions have to be bound in stable complexes which should contain at least one water molecule coordinated directly to the metal ion centre. This metal-bound water molecule is responsible for relaxation enhancement of the ^1^H MRI signal due to its exchange with the bulk water which is observed by MRI [11].

Almost all MRI CAs utilized in clinical practice till now have been based on Gd(III). Their safety was challenged after observation of nephrogenic systemic fibrosis (NSF). The disease has been connected with a use of Gd(III) chelates with several open-chain ligands, namely bis(amide) derivatives of H_5_dtpa [12,13,14] and they were withdrawn from the market. Therefore nowadays, a utilization of more stable macrocyclic chelates is strongly preferred (e.g., H_4_dota, Figure 1, and its derivatives and analogues). In vivo stability of the MRI CAs is given by thermodynamic stability of the complexes and, mainly, by their kinetic inertness, and both properties have been intensively studied.

Therefore, alternatives to the Gd-based MRI CAs have been sought. A significant effort has been paid to investigations of Mn(II)-based CAs as Mn(II) is an essential biogenic metal ion (serving e.g., as a cofactor in a number of enzymes) [15] and it has a lower toxicity if compared to Gd(III). The relatively low toxicity of Mn(II) even enables use of simple salts, such as MnCl_2_, in the specific imaging field called manganese enhanced MRI (MEMRI) [16] which allows detailed study of brain structure and local brain/neuronal or cardiac function due to similarity of Mn(II) ionic radius with that of Ca(II) and its ability to substitute this ion in biological processes. However, it is limited only to experimental work with small animals as Mn(II) is anyway toxic at concentrations needed to provide a sufficient contrast (LD_50_(MnCl_2_) = 0.22 mmol kg^−1^ in rat) [17]. In human medicine, Teslascan^®^ ([Mn(dpdp)]^4–^, where dpdp^6–^ = *N*,*N*’-dipyridoxyl-ethylenediamine-*N*,*N*’-diacetate- 5,5′-bis(phosphate)) (Figure 1) [18], was approved for liver, kidney, and cardiac imaging [19,20]. This compound does not directly coordinate water molecules and its effectiveness lies in a slow release of free Mn(II)-aqua complex from the chelate (its slow dissociation restrains acute toxicity potentially inflicted by a direct application of the same dose as e.g., MnCl_2_). However, accumulation of Mn(II) in the brain has been proven [21], and overexposure to Mn(II) can lead to neurological disorders resulting in a form of parkinsonism termed manganism [22]. Therefore, careful ligand design still remains a crucial point also in the area of the Mn(II)-based CAs.

Other important field of utilization of metal chelates with polydentate ligands is nuclear medicine where metal radionuclides are used. Over the last years, a number of metal radioisotopes with properties useful for both radiodiagnosis and radiotherapy have become widely available. Therefore, there is a need for (optimally macrocyclic) ligands able to selectively complex metal ions as Ga(III), Sc(III), Y(III), lanthanides(III), Cu(II), In(III), or Bi(III) [23,24,25,26,27]. These facts make research on macrocyclic ligating systems a vital part of modern coordination chemistry.

To contribute to the field, we decided to study a series of ligands based on 1,4,7,10-tetraazacyclododecane (cyclen) endowed by two coordinating pendant arms bound in its 1,4-positions as data on this family of ligands are almost missing in the literature. Only both analogous acetate derivatives (1,4- and 1,7-bis(carboxymethyl) cyclen derivatives, 1,4-H_2_do2a and 1,7-H_2_do2a, respectively, Figure 1) have been reported and studied [28,29,30,31,32,33]. Among their phosphorus acid analogues, only isomeric 1,7-bis(methylenephosphonic acid) derivative (1,7-H_4_do2p, Figure 1) and its bis(monoethyl-ester) (1,7-H_2_do2p^OEt^, Figure 1) have been reported and the Mn(II) complex of 1,7-H_4_do2p has been investigated [34]. Here we report on synthesis of three new ligands, 1,4-H_4_do2p, 1,4-H_2_do2p^OEt^, and 1,4-H_2_Bn_2_do2p^H^ (with phosphonate, phosphonate monoethyl ester, and H-phosphinate pendant moieties, Figure 1), and on their coordination chemistry with selected metal ions relevant for medicinal imaging techniques.

## 2. Results and Discussion

### 2.1. Synthesis

Synthesis of the studied ligands 1,4-H_4_do2p, 1,4-H_2_do2p^OEt^, and 1,4-H_2_Bn_2_do2p^H^, is shown in Scheme 1. The starting 1,4-dibenzyl-1,4,7,10-tetraazacyclododecane (**1**) was prepared by a slight modification of the literature method. Compared to the reported procedure [35], ethanol was added into the reaction mixture for better solubility of the starting oxalyl-protected dibenzylcyclen during its basic hydrolysis (see ESI).

Diethyl phosphonomethyl moieties were introduced by a Mannich-type reaction (under analogous conditions as those reported in literature [36]) to obtain compound **2**. Hydrogenolysis of **2** using H_2_ and Pd/C as a catalyst afforded compound **3**. Trials to fully hydrolyse this compound to the required ligand 1,4-H_4_do2p failed—acid hydrolysis (heating in aq. HCl) led to rich mixtures. However, basic hydrolysis [37,38] of the compound **3** in aq. NaOH selectively led to the bis-monoesterified ligand 1,4-H_2_do2p^OEt^. Thus, the ligand 1,4-H_4_do2p was prepared by a reversed approach. In the first step, ethyl ester groups were removed by trans-esterification with trimethylsilyl bromide and the silyl ester hydrolysis with aq. MeOH, affording intermediate **4**. Surprisingly in this step, the classical acid hydrolysis of **2**, i.e., heating in aq. HCl, did not lead to any ester hydrolysis. In the second step, benzyl groups were removed by a trans-acylation approach. Thus first, full silyl ester soluble in anhydrous media was prepared, benzyl groups were removed in reaction with benzyl chloroformate (Z-Cl) and the carbamate group was hydrolyzed in HBr/AcOH, similarly as reported in literature [39]. Hydrogenolysis of compound **4** (H_2_, Pd/C catalyst) was not successful, probably due to a low solubility of the compound **4** in several acidic solvents.

To synthesize 1,4-H_2_Bn_2_do2p^H^, reaction between **1**, hypophosphorus acid and paraformaldehyde in acid media was employed. Removal of benzyl groups in the compound either by hydrogenation (H_2_, Pd/C catalyst) or trans-acylation (using Z-Cl, as described above) was not successful as rich mixtures of various products were obtained. Therefore, the benzylated derivative 1,4-H_2_Bn_2_do2p^H^ was further studied as an example of a ligand with H-phosphinic acid pendant arms.

Tetrahydrate of 1,4-H_4_do2p was studied by single-crystal X-ray diffraction. In the solid state, the molecule has a zwitterionic structure where each phosphonate pendant moiety is singly protonated and the other two protons are located on mutually trans macrocycle amino groups (Figure 2). Both pendant arms point to the same half-space with respect to the macrocycle ring and are mutually interacting through a strong intramolecular hydrogen bond (*d*_O11∙∙∙O22_ = 2.60 Å). Intramolecular hydrogen bonds involving the macrocycle amino groups stabilize the ring in (3,3,3,3)-B conformation [40] (Appendix A). All four water molecules of crystallization participate in a wide intermolecular hydrogen bond network (Appendix A).

### 2.2. Protonation of the Ligands and Stability of Their Complexes

Stepwise protonation constants of 1,4-H_4_do2p (H_4_L1), 1,4-H_2_do2p^OEt^ (H_2_L2), and 1,4-H_2_Bn_2_do2p^H^ (H_2_L3) and the stability constants of their complexes with selected metal ions were determined by standard potentiometric titrations. A full set of the experimental values (with standard deviations) is outlined in the Supporting Information (Appendix A).

The protonation constants of the ligands are listed in Table 1. Their values indicate that the phosphonate derivative, 1,4-H_4_do2p, has the highest ring amine group basicity among the ligands except the first H_4_dota constant [41]. The values of 1,4-H_4_do2p are similar to those of its isomer, 1,7-H_4_do2p [34]. It is expectable as the doubly negatively charged phosphonate groups exhibit +*I* effect and it has been previously observed for other ligands with phosphonate pendant arms [42]. In addition, the ring amine group basicity can be also increased due to a strong hydrogen bond between the amine group(s) and the pendant arm [39]. On the other hand, deprotonated phosphonate-monoester and phosphinate groups having −*I* effect decrease basicity of the neighboring ring amine groups, and they do not form strong intramolecular hydrogen bonds with the protonated amine groups [42]. Consequently, the first protonation constants, log*K*_H1_, of 1,4-H_2_do2p^OEt^, 1,4-H_2_Bn_2_do2p^H^, and the related 1,7-H_2_do2p^OEt^ [34] are more than one order of magnitude lower than the corresponding values for the phosphonate derivatives. The second protonation constants, log*K*_H2_, of 1,4-H_2_do2p^OEt^ and 1,4-H_2_Bn_2_do2p^H^ are lower than those found for the acetate derivatives and the lowest value was found for 1,4-H_2_Bn_2_do2p^H^ containing *N*-benzyl groups which further decrease basicity of the amine groups. The third and fourth protonation constants of 1,4-H_4_do2p obviously correspond to the second protonation step of the phosphonate pendant moieties and the values are similar to those of its 1,7-isomer [34]. Values of these constants lie in the range of values commonly found for the groups [42,43,44,45]. For 1,4-H_4_do2p and, 1,4-H_2_do2p^OEt^, the other protonations occurring in a strong acid region (log*K*_H5_ and log*K*_H3_, respectively) were found. They could be assigned to protonation of the pendant arm or further protonation of the macrocycle amino groups. Ligand distribution diagrams are shown in Appendix A.

Stability constants of the metal complexes are given in Table 2, and representative distribution diagrams of selected ligand–metal ion systems are shown in Figure 3 and Appendix A. The stability constants of the title ligands with studied transition metal ions, Mn(II), Cu(II), and Zn(II), abide the Irving–Williams order and the ligands are very selective for the Cu(II) ion (about five orders of magnitude) over Zn(II) ion. This selectivity together with a high stability of the Cu(II) complexes (quantitative complex formation at pH ~3–4, Appendix A) suggests that the ligands are potentially useful for radiopharmaceuticals employing copper radioisotopes. Another notable difference arises from comparison of stability of Mn(II) and Ca(II) complexes. The studied ligands strongly prefer complexation with Mn(II) (about six orders of magnitude higher values for Mn(II)). However, Mn(II) is fully complexed only at pH ~8 (Figure 3A–C) and, thus, the complexes cannot be used in vivo.

The Mg(II) complexation was observed only for 1,4-H_4_do2p; for the other two ligands, the complexes are too weak and precipitation of the metal hydroxide occurred in the neutral region. The 1,4-H_4_do2p showed a relatively high stability of its Gd(III) complex (log *K*_LM_ = 19.15) resulting in the full complex formation at pH ~6 (Figure 3D). In the systems with the other two ligands, gadolinium(III) hydroxide precipitated before the complex formation. The stability constant of Gd(III) complex with hexadentate 1,4-H_4_do2p (log*K*_LM_ = 19.15) is lower than those of heptadentate H_3_do3a (log*K*_LM_ = 21.0) [46] or H_6_do3p (log*K*_LM_ = 24.6) [43]. It is a result of the higher denticity and basicity of these heptadentate ligands. Comparing with isomeric ligand 1,7-H_4_do2p (log*K*_LM_ = 18.2) [34], the stability of the 1,4-isomer complex is higher, probably due to the presence of a formally “ethylenediamine-like” motif with two closely located phosphonate groups.

However, a direct comparison of values of the stability constants is misleading as the ligands have different proton affinities (protonation of the donor sites is competing process to the complexation of metal ions). Therefore, concentrations of the free metal ions in the solutions with M:L molar ratio 1:1 (C_L_ = C_M_ = 0.001 M) at physiological pH were calculated, and the values are given in Table 3 together with the literature data for other ligands obtained under similar conditions.

The stability of the Gd(III)–1,4-H_4_do2p complex is slightly higher than the that of the complex of the 1,7-isomer [34]. However, it should be noted, that the thermodynamically sufficiently stable Gd(III)–H_3_do3a complex, showing a pGd value almost three orders of magnitude higher than that for the Gd(III)–1,4-H_4_do2p complex (Table 3), is not sufficiently kinetically inert to be used in vivo [50,51]. Among Mn(II) complexes with hexadentate ligands, complexes of the acetate derivatives exhibit better thermodynamic stability at neutral pH. However, all complexes show a low relaxivity due to their poor hydration (*q* < 1, see below) [31] and, mostly, they have insufficient stability as full complexation is reached at pH higher than the physiological one [32,33,34]. Increase in stability can be realized with more coordinating pendant arms (i.e., with the ligand having seven or eight donor atoms, e.g., H_3_do3a and H_4_dota). Although all donor groups of such ligands are not utilized for the metal ion coordination, stability of their complexes steeply increases as the pendant arms are probably fluxional exchanging coordinated/uncoordinated pendant groups and, thus, easily keep metal ion hexa- (N_4_O_2_) coordinated in the ligand cavity.

Generally, investigation of the thermodynamic properties of the title ligands and their complexes fill up the gap in knowledge of such macrocyclic systems, and the data follows the expected trends [42].

### 2.3. Hydration Number of Mn(II) and Gd(III) Complexes

A key parameter for MRI applications of paramagnetic complexes is undoubtedly a number of directly coordinated water molecules [52]. Although the Mn(II) and Gd(III) complexes of the studied ligands do not show sufficient stability for in vivo applications, determination of hydration of the complex species will shed some light on their molecular structure in solution and will bring potentially useful information for a design of better ligands having a higher denticity (and expected higher stability). Therefore, relaxometric data were acquired for the Gd(III)–1,4-H_4_do2p system and for Mn(II) complexes of all studied ligands. The efficiency of a paramagnetic contrast agent is commonly described by proton relaxivity, *r*_1_. It is mostly determined by a measurement of relaxation enhancement of water protons in the presence of paramagnetics in the solution at various external magnetic fields. The resulting ^1^H NMRD (nuclear magnetic relaxation dispersion) profiles are influenced by some microscopic parameters governing the relaxivity. The shape of the ^1^H NMRD profile depends on a combination of several physico-chemical parameters and can help to estimate some MRI-related properties of the paramagnetic complexes in solution. Graphical representations of obtained data are shown in Figure 4 and Appendix A, and proton relaxivity data are summarized in Table 4 together with comparison with related systems.

The ^1^H NMRD profiles measured for Mn(II)–1,4-H_4_do2p and Gd(III)–1,4-H_4_do2p systems at 25 and 37 °C are typical for low-molecular-weight paramagnetic chelates, showing a single dispersion between 1 and 10 MHz (Figure 4). In the case of the Mn(II)–1,4-H_4_do2p complex, absence of the second dispersion at low field clearly shows [7,8,53] that no free aqua Mn(II) ion is present in the solution as suggested from the distribution diagram (Figure 3). Ligands denticities (6) and the usual coordination number of Mn(II) ions (6–7) allow for a possible hydration number in a range of *q* 0–1 in these systems. The relaxivity values of the Mn(II) complexes (Figure 4 and Appendix A, Table 4) at high fields are low (in the range of *r*_1_ 1–2 mm^−1^ s^−1^) which points to a fact that no directly Mn(II)-coordinated water molecule is present (*q* = 0) [30]. Then, such a small relaxivity can be attributed to the second- and/or outer-sphere contributions.

Contrarily, the relaxivity (Figure 4, Table 4) exhibited by the Gd(III)–1,4-H_4_do2p system hints at a higher hydration number of the complex. According to the hexadentate nature of 1,4-H_4_do2p and common coordination numbers of Gd(III) ions (8–9), one would expect up to 2–3 directly coordinated water molecules to the metal centre. Considering a relatively high steric hindrance of the phosphonate groups [55], the hydration probably cannot be reached. In addition, the relaxivity value (Table 4) is not high enough to justify the presence of three directly bound water molecules, as relaxivity is only slightly higher than that for [Gd(H_2_O)_2_(do3a)] which binds two water molecules in the inner sphere [56]. Due to high hydrophilicity of the phosphonate groups, it can be suggested that some part of the overall relaxivity of the Gd(III)–1,4-H_4_do2p complex can be attributed to a second-sphere contributions and, then, *q* = 1–2 can be expected for the system. Anyway, hydration equilibrium with *q* = 2–3 cannot be fully excluded as not fully convincing hydration data were reported for the Ln(III)–1,7-H_4_do2p complexes [57].

To obtain a more reliable value of *q*, the hydration state of the central Ln(III) ions in the Eu(III) and Tb(III) complexes was probed using luminescence lifetime measurements as the lifetimes are sensitive to the presence of an O–H bond(s) as a vibrational quencher [58,59,60]. Lifetimes of the Eu(III)/Tb(III)–1,4-H_4_do2p systems measured in H_2_O and D_2_O are outlined in Table 5. From the data, the hydration number can be elucidated by well-established empirical formulas [58,59,60]. Thus, the break in hydration occurs at around Gd(III) and *q* should be in the range of 1–2 in the Gd(III)–1,4-H_4_do2p complex and it is consistent with the above suggestion based on the proton relaxivity values.

To solve the “hydration” problem, absorption spectra of the ^5^D_0_←^7^F_0_ transition for the Eu(III)–1,4-H_4_do2p complex were recorded at various temperatures. Since the transition is extremely sensitive to local coordination environment of the central Eu(III) ion, any change in hydration (i.e., in a coordination number) results in appearance of two spectral bands with peak separation larger than 0.5 nm [56,61]. Two peaks (Figure 5) with separation ~0.6 nm were observed pointing to a presence of two distinct species with the different hydration states. Abundance of each species can be determined by deconvolution (Table 6) of the absorption spectra. In general, the species with a higher hydration number is preferred at low temperature whereas increase in temperature leads to species with a lower *q* [56]. Based on the emission absorption data, it can be concluded that two species present in solution are [Eu(1,4-do2p)(H_2_O)]^−^ and [Eu(1,4-do2p)(H_2_O)_2_]^−^ complexes, with a high preference for the bis(hydrated) species at room temperature. Therefore, the [Gd(1,4-do2p)(H_2_O)_1–2_]^−^ equilibrium can be expected to take place in solution of the Gd(III)–1,4-H_4_do2p complex.

## 3. Materials and Methods

Dry solvents were prepared by standard purification procedures [62], distilled under argon, and stored over 4Å molecular sieves in argon atmosphere: tetrahydrofuran (THF, obtained from Penta, Prague, the Czech Republic, distilled from Na), *N*,*N*-dimethylformamide (DMF, obtained from Penta, Prague, the Czech Republic, distilled from P_2_O_5_), MeCN (Penta, Prague, the Czech Republic, distilled from P_2_O_5_). Other solvents and chemicals were purchased from commercial sources and used as received.

### 3.1. General

1,4-Dibenzyl-1,4,7,10-tetraazacyclododecane (**1**) was prepared according to literature [35] with slight modifications made during the scale-up process (for details, see ESI). NMR spectra were recorded at 25 °C on a VNMRS300 spectrometer (Varian Inc., Palo Alto, CA, USA): ^1^H 299.94 MHz, *t*-BuOH (internal) δ 1.25 ppm, TMS (internal) δ 0.00 ppm, CHD_2_OD (internal) δ 3.31 ppm; ^13^C 75.42 MHz, *t*-BuOH (internal) δ 30.29, and 70.36 ppm, TMS (internal) δ 0.00 ppm, CHCl_3_ (internal) δ 77.00 ppm; ^31^P 121.4 MHz, 85% H_3_PO_4_ in D_2_O (external) δ 0.00 ppm. Multiplicity of the signals is indicated as follows: s—singlet, d—doublet, t—triplet, q—quartet, m—multiplet, br—broad. The NMR samples for the compound characterizations were prepared by just dissolving of the compounds in H_2_O/D_2_O and pH/pD was not further controlled. Deuterated solvents D_2_O (99.9% D), CDCl_3_ (99.8% D), and CD_3_OD (99.8% D) purchased from Chemtrade were used as received. Mass spectra were measured on a spectrometer ESQUIRE 3000 (Bruker, Billerica, MA, USA) equipped with an electro-spray ion source and ion-trap detector in both negative and positive modes. For thin layer chromatography (TLC), aluminum foils with silica gel impregnated with a fluorescent dye were used (60 F_254_, Merck, Darmstadt*,* Germany); the compounds were visualized by UV lamp (λ 254 nm), 0.5% ninhydrin solution in EtOH, 5% aq. copper(II) acetate solution, or iodine vapors. Elemental analyses were done at the Institute of Macromolecular Chemistry (Academy of Science of the Czech Republic, Prague, Czech Republic).

### 3.2. Synthesis

*1,4-bis{[(diethoxy)phosphoryl]methyl}-7,10-dibenzyl-1,4,7,10-tetraazacyclododecane, **2**.* 1,4-dibenzyl-1,4,7,10-tetraazacyclododecane (**1**, 3.30 g, 9.38 mmol) was dissolved in triethyl phosphite (25 mL, 140 mmol), and paraformaldehyde (1.32 g, 37.5 mmol) was added. Reaction mixture was heated at 30 °C and vigorously stirred. Reaction progress was followed by TLC (NH_3_:EtOH 1:10, **1**
*R*_f_ 0.4; **2**
*R*_f_ 0.8). When no compound **1** was present on TLC (usually after 24 h), the reaction mixture was filtered and loaded on a strong cation exchange resin (Dowex 50, 400 cm^3^, H^+^-cycle, rinsed by EtOH). The triethyl phosphite excess was removed by EtOH (700 mL), other impurities were removed by EtOH:H_2_O (1:1 v/v, 700 mL), and the product was eluted by aq. conc. NH_3_:EtOH (1:5, 1000 mL). The product **2** was isolated as a yellowish oil after evaporation of the solvents (4.8 g, 77% based on **1**).

NMR (CDCl_3_): ^1^H *δ* 1.29 (CH_3_, 12H, t, ^3^*J*_HH_ = 7.20 Hz); 2.60–2.78 (N-CH_2_-CH_2_-N, 8H, br); 2.80–3.00 (N-CH_2_-C*H*_2_-N, N-CH_2_-P, 12H, br); 3.48 (N-CH_2_-Ph, 4H, s); 4.01 (O-CH_2_-CH_3_, 8H, m); 7.19–7.30 (arom, 10H, m); ^13^C{^1^H} *δ* 16.5 (*C*H_3_, d, ^3^*J*_PC_ = 5.5 Hz); 50.0 (N-*C*-P, d, ^1^*J*_PC_ = 153 Hz); 52.2, 52.3, 53.5 (^3^*J*_PC_ = 6.8 Hz), and 53.6 (^3^*J*_PC_ = 6.8 Hz) (all N-C-C-N); 60.1 (N-*C*-Ph, s); 61.6 (O-*C*H_2_-CH_3_, d, ^2^*J*_PC_ = 7.2 Hz); 126.7 (arom.); 128.0 (arom.); 129.1 (arom.); 139.6 (arom., quarternary); ^31^P{^1^H} *δ* 26.1 (s); **^31^**P *δ* 26.1 (m). ESI-MS *m*/*z* (+) found: 653.3 (100%); 654.3 (39%); calcd.: [M + H]^+^, C_32_H_55_N_4_O_6_P_2_^+^, including isotopic pattern: 653.4 (100%) and 654.4 (35%). TLC: conc. aq. NH_3_:EtOH 1:10, *R*_f_ 0.8.

*1,4-bis{[(diethoxy)phosphoryl]methyl}-1,4,7,10-tetraazacyclododecane, **3**.* Compound **2** (4.7 g, 7.2 mmol) was dissolved in aq. acetic acid (1:1 v/v, 50 mL) and catalyst (10% Pd/C, 0.8 g) was added. Balloon with H_2_ was attached to the flask. Reaction progress was followed by TLC (NH_3_:EtOH 1:10; **2**
*R*_f_ 0.8; **3**
*R*_f_ 0.4). After 1 day, the balloon was refilled. Next day, the catalyst was filtered off and the filtrate was co-evaporated with water and afterwards several times with ethanol to remove volatiles. Product (3.2 g, 94%) was isolated as a pale yellow oil.

NMR (D_2_O): ^1^H *δ* 1.27 (CH_3_, 12H, t, ^3^*J*_HH_ = 6,90 Hz); 2.86–3.35 (N-CH_2_-CH_2_-N, N-CH_2_-P, 20H, br); 4.15 (O-CH_2_-CH_3_, 8H, m); ^13^C{^1^H} *δ* 18.2 (*C*H_3_, d, ^3^*J*_PC_ = 5.5 Hz); 45.0; 49.1; 51.1; 52.3 (N-C-C-N, s); 53.2 (N-*C*H_2_-P, d, ^1^*J*_PC_ = 146 Hz); 61.6 (O-*C*H_2_-CH_3_, d, ^2^*J*_PC_ = 7.2 Hz). ^31^P{^1^H} *δ* 29.7 (s); ^31^P *δ* 29.7 (m). ESI-MS *m*/*z* (+) found: 473.1 (100%); 474.1 (20%); calcd.: [M + H]^+^, C_18_H_43_N_4_O_6_P_2_^+^, including isotopic pattern: 473.3 (100%) and 474.3 (20%). TLC: conc. aq. NH_3_:EtOH 1:10, *R*_f_ 0.4; conc. aq. NH_3_:EtOH 1:5, *R*_f_ 0.6.

*1,4-bis{[(ethoxy)(hydroxyl)phosphoryl]methyl}-1,4,7,10-tetraazacyclododecane, 1,4-H_2_do2p^OEt^*. Compound **3** (1.3 g, 2.0 mmol) was dissolved in ethanol (20 mL) and a solution of NaOH (0.4 g, 10 mmol) in water (10 mL) was added. Reaction progress was followed by TLC (NH_3_:EtOH 1:5; **3**
*R*_f_ 0.6; 1,4-H_2_do2p^OEt^
*R*_f_ 0.3). After 24 h, reaction mixture was evaporated, the residue was dissolved in water (5 mL), loaded onto a silica column (200 mL), and chromatographed with NH_3_:EtOH 1:5. Fractions containing product (TLC check) were combined, evaporated, and loaded onto a weak cation exchange resin (Amberlite 50, 150 mL, H^+^-cycle). The column was eluted with water. Fractions containing the pure ligand 1,4-H_2_do2p^OEt^ were combined and evaporated to a colorless oil which affords a transparent glassy material after lyophilization (1.02 g, 90%).

NMR (D_2_O): ^1^H *δ* 1.15 (CH_3_, 6H, t, ^3^*J*_HH_ = 7.20 Hz); 2.95–3.05 (N-CH_2_-CH_2_-N, 7H, br); 3.05–3.28 (N-CH_2_-CH_2_-N, N-CH_2_-P, 13H, br); 3.88 (O-CH_2_-CH_3_, 4H, m); ^13^C{^1^H} *δ* 18.5 (*C*H_3_, d, ^3^*J*_PC_ = 5.0 Hz); 44.5; 45.8 (N-C-C-N); 51.4 (N-C-P, d, ^1^*J*_PC_ = 144 Hz); 53.2; 54.3 (C-N-C-P); 63.4 (*C*H_2_-CH_3_, d, ^2^*J*_PC_ = 6.0 Hz); ^31^P{^1^H} *δ* 17.7 (s); ^31^P *δ* 17.7 (br); ESI-MS *m*/*z* found: (−) 414.9 (100%); 415.8 (17%); (+) 439.0 (31%); 461.0 (37%); 477.0 (100%); calc.: (−) [M − H]^−^, C_14_H_33_N_4_O_6_P_2_^−^, including isotopic pattern: 415.2 (100%) and 416.2 (15%); (+) [M + Na]^+^, C_14_H_34_N_4_NaO_6_P_2_^+^: 439.2; [M − H + 2Na]^+^, C_14_H_33_N_4_Na_2_O_6_P_2_^+^: 461.2; [M − H + Na + K]^+^, C_14_H_33_KN_4_NaO_6_P_2_^+^: 477.1. Elem anal.: found (calcd. for 1,4-H_2_do2p^OEt^∙4H_2_O, C_14_H_42_N_4_O_10_P_2_): C 34.59 (34.43); H 8.72 (8.67); N 11.37 (11.47). TLC: conc. aq. NH_3_:EtOH 1:5, *R*_f_ 0.3.

*1,4-bis{[(dihydroxy)phosphoryl]methyl}-7,10-dibenzyl-1,4,7,10-tetraazacyclododecane, **4.*** Compound **2** (2.30 g, 3.53 mmol) was dissolved in dry MeCN (40 mL) and trimethylsilyl bromide (10.8 g, 70.5 mmol) was added. The solution was stirred at room temperature for 18 h, MeOH (20 mL) was added, and solvents were evaporated off in vacuum. The residue was dissolved in MeOH (50 mL) and the solvent was evaporated in vacuum, and the co-evaporation with MeOH was repeated two more times. The crude product was purified on strong cation exchange resin (Dowex 50, 300 mL, H^+^-cycle). The column was eluted by water (1000 mL) followed by conc. aq. NH_3_:EtOH mixture (1:5 v/v, 700 mL). Product was eluted in the ammonia phase and it was obtained as yellowish powder after evaporation of the solvents (1.80 g) which was directly used in the next step.

NMR (D_2_O): ^1^H δ 2.95–3.40 (N-CH_2_-CH_2_-N, N-CH_2_-P, 20H, br); 3.88 (N-CH_2_-Ph, 4H, s); 7.20–7.40 (arom., 10H, m); ^13^C{^1^H} δ 49.3; 51.7 (N-C-C-N); 52.8 (N-C-P, d, ^1^*J_PC_* = 153 Hz); 52.9; 53.9 (N-C-C-N); 60.1 (N-C-Ph, s); 131.9 (arom.); 132.3 (arom.); 132.9 (arom.); 133.5 (arom.); ^31^P{^1^H} δ 16.9 (bs); ^31^P δ 16.9 (br); ESI-MS *m/z* found: (−) 539.0 (100%); 540.0 (24%); (+) 541.1 (36%); 563.1 (100%); 579.1 (68%); calc.: (−) [M − H]^−^, C_24_H_37_N_4_O_6_P_2_^−^, including isotopic pattern: 539.2 (100%) and 540.2 (26%); (+) [M + H]^+^, C_24_H_39_N_4_O_6_P_2_^+^: 541.2; [M + Na]^+^, C_24_H_38_N_4_NaO_6_P_2_^+^: 563.2; [M + K]^+^, C_24_H_38_KN_4_O_6_P_2_^+^: 579.2. TLC: conc. aq. NH_3_:EtOH 1:5, *R*_f_ 0.4.

*1,4-bis{[(dihydroxy)phosphoryl]methyl}-1,4,7,10-tetraazacyclododecane, 1,4-H_4_do2p.* Compound **4** (1.8 g, 3.4 mmol) was suspended in mixture of toluene (20 mL) and anhydrous THF (10 mL) and the solvents were evaporated in vacuum. The pre-dried compound **4** was suspended in anhydrous THF (40 mL) and hexamethyldisilazane (20 mL) was added. The mixture was heated to reflux for 1 h. The mixture was evaporated to dryness in a flow of dry argon and the residue was dissolved in anhydrous THF (40 mL). Benzyl chloroformate (5.0 g) was added and the mixture was stirred for 24 h. Then, mixture of conc. aq. NH_3_ (5 mL) and EtOH (5 mL) was added and the mixture was stirred for 1 h. The reaction mixture was evaporated to dryness in vacuum, the residue was dissolved in HBr/AcOH (30% w/w) mixture and it was stirred at room temperature for 1 h. Afterwards, the mixture was evaporated to dryness, the remaining material was dissolved in water (30 mL) and the solution was washed three times with CHCl_3_ (20 mL). The aqueous phase containing the crude product was purified on strong cation exchange resin (Dowex 50, 300 mL, H^+^-cycle). Column was eluted by water (1000 mL) and 5% aq. NH_3_ (700 mL). The ammonia fraction was evaporated to dryness giving a crude product. It was further purified on a weak cation exchange resin (Amberlite CG50, 300 mL, H^+^-cycle) using water as an eluent. Fractions containing pure product were combined and evaporated to dryness. The product, 1,4-H_4_do2p, was obtained as a white powder upon evaporation to dryness (1.04 g, 87% over two steps). Single crystals suitable for X-ray diffraction study were obtained upon slow evaporation of aq. solution.

NMR (D_2_O): ^1^H δ 3.00–3.18 (N-CH_2_-CH_2_-N, 12H, m); 3.24 (N-CH_2_-P, 4H, d, ^2^*J_PH_* = 7.80 Hz); 3.30 (N-CH_2_-CH_2_-N, 4H, s); ^13^C{^1^H} δ 44.4; 45.1 (N-C-C-N); 52.7 (N-C-P, d, ^1^*J_PC_* = 145 Hz); 53.8; 53.9 (C-N-C-P); ^31^P {^1^H} δ 16.9 (bs); ^31^P δ 16.9 (br m); ESI-MS *m/z* found: (−) 358.7 (100%); 359.7 (12%); (+) 360.8 (100%); 398.8 (28%); calc.: (−) [M − H]^−^, C_10_H_25_N_4_O_6_P_2_^−^, including isotopic pattern: 359.1 (100%) and 360.1 (11%); (+) [M + H]^+^, C_10_H_27_N_4_O_6_P_2_^+^: 361.1; [M + K]^+^, C_10_H_26_KN_4_O_6_P_2_^+^: 399.1. Elem anal.: Found (calcd. for 1,4-H_4_do2p∙2.5H_2_O, C_10_H_31_N_4_O_8.5_P_2_): C 29.79 (29.63); H 7.13 (7.71); N 13.71 (13.82). TLC: ^i^PrOH: conc. aq. NH_3_:H_2_O 7:3:3, *R*_f_ 0.6.

*1,4-dibenzyl-7,10-bis{[(hydroxyl)(hydrido)phosphoryl]methyl}-1,4,7,10-tetraazacyclododecane, 1,4-H_2_Bn_2_do2p^H^.* Compound **1** (0.640 g, 1.82 mmol) was dissolved in mixture of EtOH (5 mL) and aq. HCl (1:1, 5 mL). Paraformaldehyde (0.32 g, 10.6 mmol) and aq. H_3_PO_2_ (50%, 3.8 mL, 28.4 mmol) were added. The reaction mixture was heated at 50 °C for 15 h. The mixture was evaporated and the residue was loaded onto a strong cation exchange resin (Dowex 50, 250 mL, H^+^-cycle). The column was eluted by water (1000 mL) and a mixture of aq. conc. NH_3_:EtOH (1:5, 700 mL). The ammonia phase was evaporated to dryness affording pure product as white powder (0.77 g, 84%).

NMR (D_2_O): ^1^H δ 2.60–3.22 (N-CH_2_-CH_2_-N, N-CH_2_-P, 20H, m); 3.74 (Ph-CH_2_-N, 4H, s); 7.02 (P-H, 2H, partly exchanged with the solvent, d, ^1^*J_PH_* = 517 Hz); 7.12–7.40 (arom., 10H, m); ^13^C{^1^H} δ 50.6; 52.2 (N-C-C-N, s); 52.9 (C-N-C-P, d, ^2^*J_PC_* = 4.5 Hz); 54.4 (N-C-C-N, s); 56.0 (N-C-P, d, ^1^*J_PC_* = 93 Hz); 59.8 (N-C-Ph); 131.5 (arom.); 131.6 (arom.); 133.3 (arom.); 134.6 (arom., quarternary); ^31^P{^1^H} δ 19.6 (br); ^31^P δ 19.6 (d, br, ^1^*J_PH_* = 517 Hz); ESI-MS *m/z* found: (−) 507.0 (100%); 507.9 (30%); (+) 509.1; 531.1; 553.1; calc.: (−) [M − H^+^]^−^, C_24_H_37_N_4_O_4_P_2_^−^, including isotopic pattern: 507.2 (100%) and 508.2 (26%); (+) [M + H]^+^, C_24_H_39_N_4_O_4_P_2_^+^: 509.2; [M + Na]^+^, C_24_H_38_N_4_NaO_4_P_2_^+^: 531.2; [M − H + 2Na]^+^, C_24_H_37_N_4_Na_2_O_4_P_2_^+^: 553.2. TLC: conc. aq. NH_3_:EtOH 1:5, *R*_f_ 0.5.

### 3.3. Potentiometric Titrations

Potentiometric titrations were carried out to determine protonation constants of the ligands and stability constants of their complexes with selected metal ions at 1:1 metal-to-ligand molar ratio. Titrations were performed at 25.0 ± 0.1 °C and at an ionic strength of *I* = 0.1 M (tetramethylammonium chloride) using deionized water. The constant passage of argon saturated with the solvent vapour provided the inert atmosphere. The initial volume in the titration vessel was ~5 mL (1,4-H_4_do2p, 1,4-H_2_do2p^OEt^) or ~3 mL (1,4-H_2_Bn_2_do2p^H^). The titrations were performed with tetramethylammonium hydroxide solution (~0.2 M) and at the ligand concentration ~0.004 M (1,4-H_4_do2p, 1,4-H_2_do2p^OEt^) or ~0.003 M (1,4-H_2_Bn_2_do2p^H^). For each system, at least three parallel titrations were carried out, each titration consisting of about 50 points. All equilibria were established quickly except the Gd^3+^–1,4-H_4_do2p system. In this case, the out-of-cell method was applied using sealed glass ampoules with a waiting time of two weeks for the equilibration.

For the ligand and the ligand–metal ion systems, titrations were run in the −log[H^+^] range 1.7–12.1 and 2.0–11.5 (or until precipitation of the metal hydroxide), respectively, employing a PHM 240 pH-meter, a 2-mL ABU 901 automatic piston burette, and a GK 2401B combined electrode (all Radiometer, Copenhagen, Denmark). The OPIUM software package was used for the data treatment [63,64]. The value of p*K*_w_ used in calculations was 13.81. Stability constants of the M^2+^-OH^−^ systems were taken from literature [65,66,67]. For more details about potentiometric titrations, see previous papers [68,69]. In the following text, pH will mean −log[H^+^] and the equilibrium constants are concentration constants.

### 3.4. Proton Nuclear Magnetic Relaxation Dispersion (^1^H NMRD) Measurements

The ^1^H NMRD profiles of aq. solutions of the Mn(II) and Gd(III) complexes (C_MnL_, C_GdL_ = 5 mm) with pH 9.5 (Mn(II)–1,4-H_4_do2p), 9.0 (other Mn(II) complexes), or 8.5 (Gd(III)–1,4-H_4_do2p) were measured at 25 and 37 °C on a SMARTracer Fast Field-Cycling NMR relaxometer (Stelar, Mede, Italy, 0.00024–0.24 T, 0.01–10 MHz ^1^H Larmor frequency) and a WP80 NMR electromagnet (Bruker, Billerica*,* MA, USA) adapted to variable-field measurements (0.47–1.88 T, 20–80 MHz ^1^H Larmor frequency), and controlled by the SMARTracer PC–NMR console. The temperature was controlled by a VTC91 temperature control unit and maintained by a gas flow. The temperature was determined according to previous calibration with a Pt resistance temperature probe.

### 3.5. UV-Vis Spectroscopy

The UV-visible spectra of ^5^D_0_←^7^F_0_ transitions of the Eu(III)–1,4-H_4_do2p complex aqueous solution (0.0185 M, pH = 8.5) were recorded in the temperature range 5–80 °C on a Perkin-Elmer Lambda 19 spectrometer in thermoregulated cuvette with a 5-cm optical length (*λ* = 577–581 nm, data steps 0.01 nm). Deconvolution of the spectra was done by minimization Excel solver function (Excel 2003, Microsoft, Redmond, WA, USA).

### 3.6. Luminescence Spectroscopy

The luminescence measurements of the Eu(III)/Tb(III)–1,4-H_4_do2p complexes (0.1 M, pH 8.5) were performed on a FS900 spectrofluorimeter (Edinburgh Instruments, Livingston, UK), equipped with a 450 W xenon arc lamp, a microsecond flash lamp, and a red-sensitive photomultiplier (300–850 nm). The luminescence spectra were obtained after excitation at 394 and 352 nm for the Eu(III) and Tb(III) complexes, respectively. Emission lifetimes were determined by single-exponential fitting of emission intensity time trace employing peaks at 614 and 545 nm for the Eu(III) and Tb(III) complexes, respectively.

### 3.7. Single-Crystal X-Ray Analysis

Single crystals of 1,4-H_4_do2p∙4H_2_O were prepared by slow evaporation of the ligand aq. solution in a form of colorless prisms. The diffraction data were collected using a Nonius Kappa CCD diffractometer (Enraf-Nonius, Rotterdam, the Netherlands) at 150(1) K (Cryo-stream Cooler, Oxford Cryosystem, Oxford, UK) using Mo-K_α_ radiation (*λ* = 0.71073 Å) and analyzed using the HKL DENZO program package [70,71]. The structure was solved using direct methods and refined by full-matrix least-squares techniques (SHELXS97 [72,73] and SHELXL2014 [74]). Final parameters: triclinic system, space group *P*−1, *a* = 7.7439(2) Å, *b* = 10.7128(3) Å, *c* = 12.0966(4) Å, *β* (°) = 100.368(1), *U* = 957.24(5) Å^3^, *Z* = 2, 4384 total refl., 4035 obsd. refl. (*I* > 2σ(*I*)), *R*_1_ = 0.0285, *wR*_2_ = 0.0772, CCDC reference no. 1943113. All non-hydrogen atoms were refined anisotropically. The hydrogen atoms were located in the electron density map, however; those belonging to the carbon atoms were fixed in the theoretical positions using the riding model with *U*_eq_(H) = 1.2 *U*_eq_(X).

## 4. Conclusions

Three new ligands based on 1,4-disubstituted cyclen with two pendant arms containing phosphonic acid, ethyl-phosphonic acid monoester, and H-phosphinic acid functional groups were synthesized. Ligand protonation and complex stability constants with selected metal ions were determined by potentiometric titrations. The phosphonate derivative is the most basic among the ligands, in accordance with the positive inductive effect of the fully deprotonated phosphonate groups. The phosphonate ligand also forms the most stable complexes with all investigated metal ions and is the only one capable of Gd(III) complexation. Both other ligands, 1,4-H_2_do2p^OEt^ and 1,4-H_2_Bn_2_do2p^H^, form significantly weaker complexes and are not able to complex lanthanide(III) ions in respect to competing metal hydroxide precipitation. All ligands are high selective for divalent copper. The Mn(II) complexes of the ligands show low relaxivities. Therefore, there is probably not enough space for a directly coordinated water in their coordination spheres, and the relaxivity is caused only by second-/outer-sphere contributions. A combined ^1^H NMRD, UV-Vis, and luminescence study on Ln(III) complexes allowed the conclusion that the Gd(III)–1,4-H_4_do2p complex exists in aqueous solution in an equilibrium of mono- and bis-hydrated species, [Gd(1,4-do2p)(H_2_O)_1–2_]^−^. The complex shows higher relaxivity than the [Gd(do3a)(H_2_O)_2_] complex with a similar hydration state, probably due to presence of a strong second-sphere hydration induced by the phosphonate groups.

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
