# Peer review of "Coordination Behavior of 1,4-Disubstituted Cyclen Endowed with Phosphonate, Phosphonate Monoethylester, and H-Phosphinate Pendant Arms"

_molecules, 2019, doi:10.3390/molecules24183324_

Round 1

Reviewer 1 Report

The manuscript is prepared with care about the details. Overall it reads very well, with only a few minor issues which can be easily addressed prior the final acceptance:

The authors should pay more attention into a proper explanation of abbreviations used. e. g line 104 TMSBr, line 219  - paramagnetic CA, line 415 - NMe4OH etc. Please provide a more detailed list of abbreviations used. Minor spell checking is required throughout the text e.g line 224 - "physic-chemical" should rather be "phycicochemical"

In general, the manuscript is well-tailored and I gladly recommend its publication after very minor revision

Author Response

The authors should pay more attention into a proper explanation of abbreviations used. e. g line 104 TMSBr, line 219 - paramagnetic CA, line 415 - NMe4OH etc. Please provide a more detailed list of abbreviations used. Minor spell checking is required throughout the text e.g line 224 - "physic-chemical" should rather be "phycicochemical"

Manuscript was read carefully and several typos were corrected. Some abbreviations used in text were replaced by full name (when the abbreviation appeared only once in the manuscript, e.g. NMe4OH, TMSBr) or were explained when appeared for the first time in respective chapter. However, meaning of "Me" and "Et" as shorts defined by IUPAC is not explained as it is too trivial (textbook knowledge).

Reviewer 2 Report

The manuscript entitled  “Coordination Behaviour of 1,4-Disubstituted Cyclen Endowed with Phosphonate, Phosphonate Monoethylester and H-Phosphinate Pendant Arms” by JiÅ™í Bárta, Petr Hermann and Jan Kotek is an original research article suitable for publication in Molecules. I have found the paper interestingly written. The introduction is rather broad – the Authors describe possible and already found applications of macrocyclic ligands based on cyclen scaffold. They relate mainly to applications associated with MRI imaging and healthcare. In this context, I would like to underline, that for the importance of macrocyclic ligands (such as cyclen, cyclam or tacn core) they have been utilized to achieve the first heteronuclear lanthanide based system, exhibiting up-conversion in aqueous solution. This means that these scaffolds can be effectively utulized also in the field of microscopy allowing lanthanide ions to aggregate close enough to enable efficient excited energy transfer between them (e.g.  A. Nonat, S. Bahamyirou, A. Lecointre, F. Przybilla, Y. Mély, C. Platas-Iglesias, F. Camerel, O. Jeannin and L. J. Charbonnière, Journal of the American Chemical Society, 2019, 141, 1568–1576; A. Nonat, C. F. Chan, T. Liu, C. Platas-Iglesias, Z. Liu, W.-T. Wong, W.-K. Wong, K.-L. Wong and L. J. Charbonnière, Nature Communications, 2016, 7, 11978; A. Nonat, M. Regueiro-Figueroa, D. Esteban-Gómez, A. de Blas, T. Rodríguez-Blas, C. Platas-Iglesias and L. J. Charbonnière, Chemistry–A European Journal, 2012, 18, 8163–8173.; A. M. Nonat, C. Allain, S. Faulkner and T. Gunnlaugsson, Inorganic Chemistry, 2010, 49, 8449–8456. ). The cyclen scaffold has been also found to enhance detection limit in fluoride sensing e.g. T. Liu, A. Nonat, M. Beyler, M. Regueiro-Figueroa, K. Nchimi Nono, O. Jeannin, F. Camerel, F. Debaene, S. Cianférani-Sanglier, R. Tripier and others, Angewandte Chemie International Edition, 2014, 53, 7259–7263.

The synthesis and characterization of ligands and complexes is well presented, the methodology is apt and of the highest standards. Moreover, the discussion of the results is exhaustive in light of well conducted literature study. Therefore, I recommend the paper for publication in Molecules, after only minor corrections to language. Below are listed selected comments concerning the main text:

The sentence “Protonations of the Cu(II), Zn(II) and Gd(III) complexes probably takes place on the coordinated phosphonate groups”could be removed from the text of abstract since it has speculative character.

Page 4, line 123 (possibly a footnote missing). A short information about crystallographic parameters (crystal system, space group, unit cell parameters, etc.) of the X-ray determination of crystal structure of the ligand should be provided in the main text also.

The sentence: “Although stability of the Gd(III)–1,4-H4do2p complex is higher than the that of the complex of the 1,7-isomer [24], the stability is not enough high for in vivo utilizations” should be corrected/modified.

Author Response

The manuscript entitled  “Coordination Behaviour of 1,4-Disubstituted Cyclen Endowed with Phosphonate, Phosphonate Monoethylester and H-Phosphinate Pendant Arms” by JiÅ™í Bárta, Petr Hermann and Jan Kotek is an original research article suitable for publication in Molecules. I have found the paper interestingly written. The introduction is rather broad – the Authors describe possible and already found applications of macrocyclic ligands based on cyclen scaffold. They relate mainly to applications associated with MRI imaging and healthcare. In this context, I would like to underline, that for the importance of macrocyclic ligands (such as cyclen, cyclam or tacn core) they have been utilized to achieve the first heteronuclear lanthanide based system, exhibiting up-conversion in aqueous solution. This means that these scaffolds can be effectively utulized also in the field of microscopy allowing lanthanide ions to aggregate close enough to enable efficient excited energy transfer between them (e.g.  A. Nonat, S. Bahamyirou, A. Lecointre, F. Przybilla, Y. Mély, C. Platas-Iglesias, F. Camerel, O. Jeannin and L. J. Charbonnière, Journal of the American Chemical Society, 2019, 141, 1568–1576; A. Nonat, C. F. Chan, T. Liu, C. Platas-Iglesias, Z. Liu, W.-T. Wong, W.-K. Wong, K.-L. Wong and L. J. Charbonnière, Nature Communications, 2016, 7, 11978; A. Nonat, M. Regueiro-Figueroa, D. Esteban-Gómez, A. de Blas, T. Rodríguez-Blas, C. Platas-Iglesias and L. J. Charbonnière, Chemistry–A European Journal, 2012, 18, 8163–8173.; A. M. Nonat, C. Allain, S. Faulkner and T. Gunnlaugsson, Inorganic Chemistry, 2010, 49, 8449–8456. ). The cyclen scaffold has been also found to enhance detection limit in fluoride sensing e.g. T. Liu, A. Nonat, M. Beyler, M. Regueiro-Figueroa, K. Nchimi Nono, O. Jeannin, F. Camerel, F. Debaene, S. Cianférani-Sanglier, R. Tripier and others, Angewandte Chemie International Edition, 2014, 53, 7259–7263.

The referee is definitely right, that macrocyclic compounds have found very broad applications. Besides up-conversion and sensitization of analytes mentioned by referees the field of applications include e.g. optical imaging applications (luminescence microscopy, organismal imaging), magnetic influence on nanodiamond fluorescence, molecular magnets, ion separations, water treatment, mining, etc. We used only some illustrative references closely related to the main topic of the article (and using relevant methodology), and we suggest do not add references regarding to other fields as it would be very expansive.

The sentence “Protonations of the Cu(II), Zn(II) and Gd(III) complexes probably takes place on the coordinated phosphonate groups”could be removed from the text of abstract since it has speculative character.

Deleted.

Page 4, line 123 (possibly a footnote missing). A short information about crystallographic parameters (crystal system, space group, unit cell parameters, etc.) of the X-ray determination of crystal structure of the ligand should be provided in the main text also.

Selected crystallographic parameters were moved from ESI into text.

The sentence: “Although stability of the Gd(III)–1,4-H4do2p complex is higher than the that of the complex of the 1,7-isomer [24], the stability is not enough high for in vivo utilizations” should be corrected/modified.

Paragraph was modified in order to remove speculations.

Reviewer 3 Report

The paper by Kotek et. al. described the coordination behaviour of 1,4-disubstituted cyclen endowed with phosphonate, phosphonate monoethylester and H-phosphinate pendant arms towards some selected metal ions relevant for medicinal imaging techniques. This is an important area of research in the development of magnetic resonance imaging contrast agents (MRICAs). Overall, this is a nice piece of work and can be accepted for publication in Molecules with some minor corrections.

There are two kinds of X-ray diffractions, single crystal and powder. So which one was used to determine the structure of the phosphonate ligand, 1,4-H4do2p should be clear in abstract section. The details about the measurement of emission lifetime should be shown. What is the NMRD? One H-bonding table should be made for 4-H4do2p∙4H2O based on the crystallographic data.

Author Response

There are two kinds of X-ray diffractions, single crystal and powder. So which one was used to determine the structure of the phosphonate ligand, 1,4-H4do2p should be clear in abstract section.

Added.

The details about the measurement of emission lifetime should be shown.

Details added.

What is the NMRD?

The short was defined in the original text, however, not in the first place when it appeared. Corrected (added).

One H-bonding table should be made for 4-H4do2p∙4H2O based on the crystallographic data.

Added to ESI.